# Computer-guided binding mode identification and affinity improvement of an LRR protein binder without structure determination

**Yoonjoo Choi**[1☯], **Sukyo Jeong**[1☯], **Jung-Min Choi**[1], **Christian Ndong**[2], **Karl E. Griswold**[2,3,4], **Chris Bailey-Kellogg**[5], **Hak-Sung Kim**[1] *

**1** Department of Biological Sciences, Korea Advanced Institute of Science and Technology, Daejeon, Korea, **2** Thayer School of Engineering, Dartmouth College, Hanover, New Hampshire, United States of America, **3** Norris Cotton Cancer Center at Dartmouth, Lebanon, New Hampshire, United States of America, **4** Department of Biological Sciences, Dartmouth College, Hanover, New Hampshire, United States of America, **5** Department of Computer Science, Dartmouth College, Hanover, New Hampshire, United States of America

☯ These authors contributed equally to this work.

* hskim76@kaist.ac.kr

**Data Availability Statement:** All relevant data are within the manuscript and its Supporting Information files.

## Abstract

Precise binding mode identification and subsequent affinity improvement without structure determination remain a challenge in the development of therapeutic proteins. However, relevant experimental techniques are generally quite costly, and purely computational methods have been unreliable. Here, we show that integrated computational and experimental epitope localization followed by full-atom energy minimization can yield an accurate complex model structure which ultimately enables effective affinity improvement and redesign of binding specificity. As proof-of-concept, we used a leucine-rich repeat (LRR) protein binder, called a repebody (Rb), that specifically recognizes human IgG$_1$ (hIgG$_1$). We performed computationally-guided identification of the Rb:hIgG$_1$ binding mode and leveraged the resulting model to reengineer the Rb so as to significantly increase its binding affinity for hIgG$_1$ as well as redesign its specificity toward multiple IgGs from other species. Experimental structure determination verified that our Rb:hIgG$_1$ model closely matched the co-crystal structure. Using a benchmark of other LRR protein complexes, we further demonstrated that the present approach may be broadly applicable to proteins undergoing relatively small conformational changes upon target binding.

## Author summary

It is quite challenging for computational methods to determine how proteins interact and to design mutations to alter their binding affinity and specificity. Despite recent advances in computational methods, however, *in silico* evaluation of binding energies has proven to be extremely difficult. We show that, in the case of protein-protein interactions where only small structural changes occur upon target binding, an integrated computational and experimental approach can identify a binding mode and drive reengineering efforts to

**Funding:** This work was supported by the Global Research Laboratory (NRF-2015K1A1A2033346) and Mid-Career Researcher Program (NRF-2017R1A2A1A05001091) through the National Research Foundation (NRF) of Korea, funded by the Ministry of Science and ICT. Y.C. was supported by the Korea Research Fellowship Program (NRF-2016H1D3A1938246) through the National Research Foundation of Korea, funded by the Ministry of Science and ICT. J.M. was supported by Basic Science Research Program (NRF-2017R1A6A3A04012313) of the National Research Foundation funded by the Ministry of Science and ICT, and the Ministry of Education. The production of the Fc and Fc variant was supported by the National Institute of General Medical Sciences of the U.S. National Institutes of Health under award numbers P20-GM113132 and 2R01GM098977. The funders had no role in study design, data collection and analysis, decision to publish, or preparation of the manuscript.

**Competing interests:** I have read the journal's policy and the authors of this manuscript have the following competing interests: Chris Bailey-Kellogg and Karl E. Griswold are Dartmouth faculties and co-members of Stealth Biologics, LLC, a Delaware biotechnology company. These authors acknowledge that there may be a potential financial conflict of interest related to their associations with this company, and they hereby affirm that the data presented in this paper are free of any bias. This work has been reviewed and approved as specified in the conflict of interest management plans of Dartmouth.

improve binding affinity or specificity. Using as a model system a leucine-rich repeat (LRR) protein binder that recognizes human IgG$_1$, our approach yielded a model of the protein complex that was very similar to the subsequently experimentally determined co-crystal structure, and enabled design of variants with significantly improved IgG$_1$ binding affinity and with the ability to recognize IgG$_1$ from other species.

## Introduction

In the development of therapeutic proteins and vaccines, the efficacy and effectiveness are largely determined by their binding modes and affinities [1–6]. Binding mode identification and affinity improvement have generally relied on labor-intensive and time-consuming experimental approaches [7], such as determination of complex structures by X-ray crystallography, and generation and screening of large libraries. To overcome these bottlenecks, considerable effort has been made to develop alternative computational methods [8, 9]. Despite some notable advances, however, computational determination of binding mode and *in silico* improvement of binding affinity remain challenging in general, and purely computational approaches have been insufficiently reliable for such purposes [10–13] and association energies [14–17]. Recent rounds of the Critical Assessment of Predicted Interactions (CAPRI) have also shown that current computational methods are not successful in identifying the actual binding mode [18]. Thus, the problem of predicting protein-protein interations is often regarded as a "Holy Grail" in the computer-aided protein engineering [14].

Leucine-rich repeat (LRR) proteins have a rigid horseshoe-like structural feature and play key roles in many biological processes [19], including the immune system [20–23] and cellular processes [24–26]. LRR proteins constitute one of the most common protein families found in a wide range of species, and more than 2,000 LRR proteins have been identified [27, 28]. Typical examples include toll-like receptors (TLR) of the mammalian innate immune system [21], and variable lymphocyte receptors (VLR) of the jawless vertebrate adaptive immune system [23]. Considering the importance and abundance of LRR proteins in nature, a broadly enabling strategy for modeling and controlling LRR binding can help in understanding of their functions as well as leveraging their recognition abilities for therapeutic applications.

We previously developed a computationally-driven epitope localization method, EpiScope, through which a target antibody's binding is evaluated against a small, optimized panel of antigenic variants to test hypothesized epitope locations [11]. EpiScope was shown to successfully predict a general epitope region, but not providing detailed information about binding mode. Here, as the extension of EpiScope, we demonstrate an integrated computational and experimental approach to identifying the binding mode that further enables affinity improvement and redesign of binding specificity. As proof-of-concept, we employed an LRR protein binder that specifically recognizes an immunoglobulin G (IgG), for which the binding mode was completely unknown. This work represents a challenging model system for affinity improvement and redesign of specificity due to high structural conservation combined with sequence diversity across species. We show that our approach effectively narrowed down the location of the binding interface, and the full-atom energy minimization identified a native-like complex model closely matching a experimentally determined X-ray crystal structure. Further computational analyses of the identified model complex allowed the design of LRR protein binders with significantly increased affinity and altered binding specificity.

## Results

### Binding mode identification of an LRR protein binder

In an effort to exploit the structural and functional features of LRR proteins for biotechnological and medical applications, we previously developed an LRR protein binder, called a "repebody (Rb)" [29]. Here, a human $IgG_1$ ($hIgG_1$)-binding repebody, named RbF4 [30], was targeted for computationally-driven binding mode identification and subsequent affinity improvement as well as redesign of binding specificity. There is indirect evidence that RbF4 recognizes the constant region of $hIgG_1$ (hFc) [30, 31], but the actual epitope residues and the binding mode were unknown. RbF4 has a typical LRR protein sequence motif, whose structural scaffold consists of three major parts: an N terminal cap (LRRNT), LRR modules (LRRVs), and a C terminal cap (LRRCT) with an additional loop (S1 Fig). In contrast to the complementarity determining regions of antibodies, the target-binding sites of a repebody (the LRRVs) comprise parallel beta strands which are assumed to remain unchanged upon target binding. During the development of RbF4, three variable residues on each of modules LRRV2, LRRV3, and LRRV5 were randomized and subjected to a phage display selection [30].

To identify the binding mode of RbF4-hFc, we first localized the RbF4 epitope on hFc using EpiScope. This computational method first predicts mutations which appear to both disrupt a target binding according to the models and maintain antigen stability; it then optimizes targeted sets of antigenic variants that combine these mutations so as to efficiently confirm and reject the various epitope hypotheses [11]. For antibody-antigen pairs, an average of three variants, each with three mutations designed to disrupt binding and maintain stability, were shown to be sufficient to test all of the docking models over a benchmark set of targets, in each case yielding at least one variant expected to include the true epitope region. It should be noted that this form of epitope localization indicates the general region where the protein is likely to bind, but does not provide a binding mode in detail.

In this study, the ClusPro webserver was employed to dock the RbF4-hFc pair as previously [11]. We used a crystal structure of the unbound form of hFc (PDB code: 3AVE) and a homology model of RbF4 for the docking (Table 1), assigning attractions at the residues of the binding site (LRRV modules from 2 to 5) in order to focus docking to this region. The affinity improvement of repebodies is similar to that of antibodies, in that diversity is generated in the binding region of the repebody/antibody followed by selection of variants with improved affinity. This leads to an assymetric representation of amino acids, and thus we used the antibody

**Table 1. Test sets.** The numbers in parentheses indicate results from ClusPro without the antibody mode option. For C5a, the numbers with asterisks (*) are results using the crystal structure for docking with the precise definition of paratopes.

| Target | | Human IgG Fc | Crystal Structure Complexes | | |
|---|---|---|---|---|---|
| | | | Interleukin 6 (IL-6) | Epidermal Growth Factor Receptor (EGFR) | Complement Component 5a (C5a) |
| Complex (PDB) | | 6KA7 | 4J4L | 4UIP | 5B4P |
| Rb homology template | | 3RFS | 5B4P | 3RFS | 4J4L |
| Unbound target | | 3AVE | 1ALU | 1NQL | 1KJS |
| Cα RMSD (Å) | Target | 1.72 | 0.92 | 2.62 | 1.76 |
| | Repebody | 1.39 | 0.63 | 1.62 | 1.17 |
| Number of docking models | | 29 | 30 (106) | 30 (88) | 23 (65) \| *28 |
| Number of EpiScope Designs | | 3 | 4 | 2 | 2 |
| Number of localized docking models | | 7 | 5 (11) | 12 (20) | 6 (34) \| *7 |
| Best I-RMSD (Å) | | 2.32 | 1.45 (1.78) | 2.23 (6.61) | 3.61 (5.53) \| *0.62 |
| Best $f_{nat}$ | | 0.43 | 0.62 (0.49) | 0.38 (0.09) | 0.27 (0.14) \| *0.76 |

docking mode and its associated assymetric scoring (Antibody Mode). As a result, a total of 29 target-bound complex models were generated (Table 1).

We note that each mutation is duplicated due to the dimeric nature of Fc. The selected variants include Var 1 (Q362E/N389E/N390K), Var 2 (H268K/E269K/R292L), and Var 3 (H310A/N315K/H435K). The binding affinities of RbF4 against the variants were experimentally evaluated by isothermal titration calorimetry (ITC), and binding of Var 3 was shown to be disrupted approximately 3-fold compared to wild-type hFc, with a decrease in $K_d$ from 128 nM to 427 nM (Fig 1B and S2 Fig). This result confirms that RbF4 indeed binds to hFc. Since, as previously observed [11], it is likely that not all positions mutated in a variant are equally important for binding, we also measured the binding affinities of the single mutations comprising Var 3. Two of the single mutations (H310A and N315K) led to meaningful two-fold reductions in binding affinity (194 and 187 nM, respectively), whereas the binding affinity of H435K

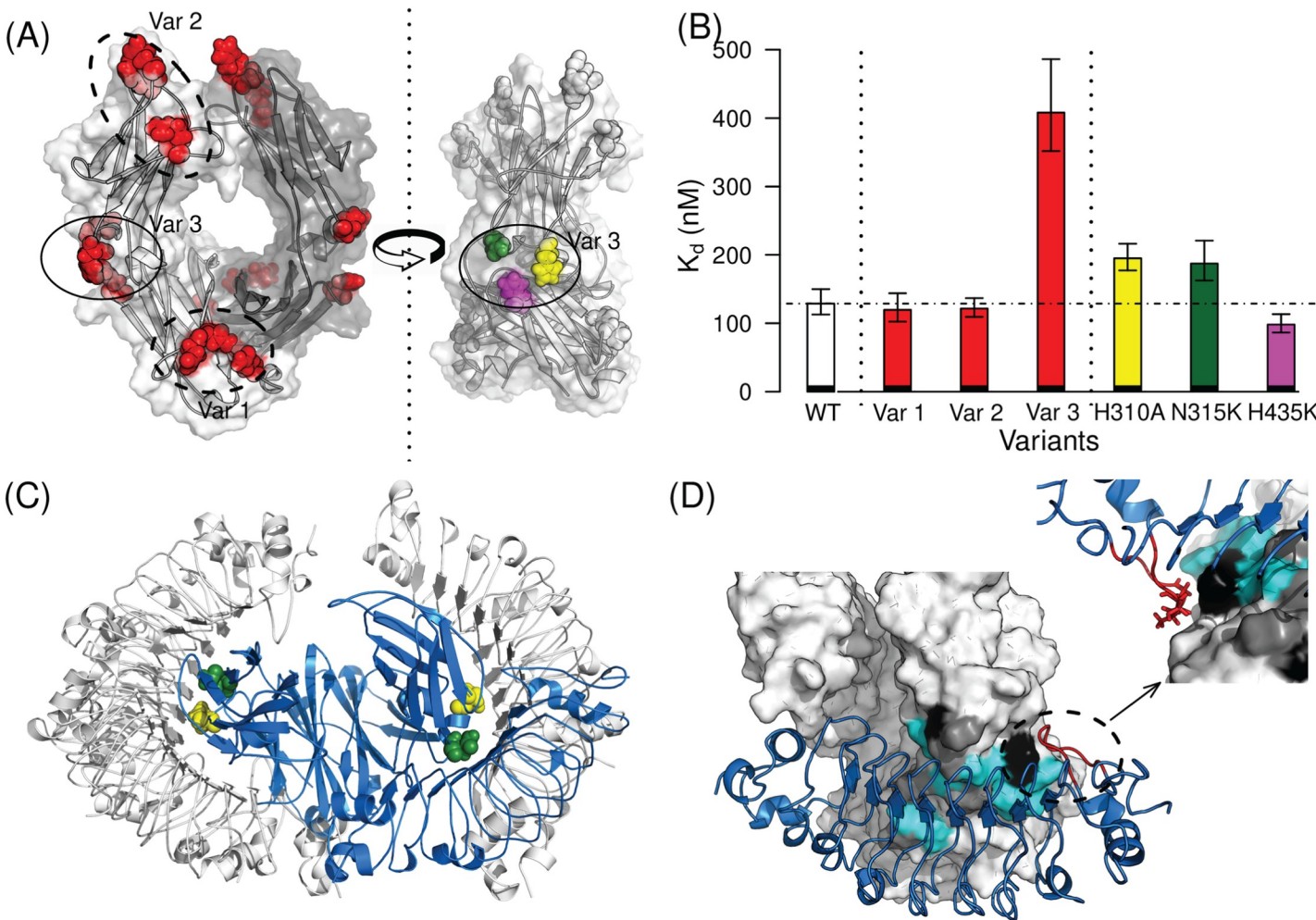

**Fig 1. Epitope localization and binding mode identification of human IgG₁ Fc (hFc)-binding repebody (RbF4).** (A) EpiScope designs triple mutants considering the symmetry of the Fc structure. (B) The set of the three mutations of Var 3 (H310A, N315K, and H435K) clearly disrupts the binding, and the single mutations comprising Var 3 were individually test. The ITC results indicate that H435K may not be involved in binding. Error bars represent variation over ITC triplicates. Details are provided in S2 Fig. (C) There are seven docking models in contact with H310A and N315K; the one colored blue had the lowest molecular modeling energy. (D) Closer inspection of the model suggests that the repebody loop (highligled in red stick; see also S1 Fig) may be responsible for the binding specificity of RbF4 to hFc. IgG from three species are considered (human; mouse, mFc; and rabbit, rFc). The residues that all three share in common are colored cyan; those common to two are gray, and those unique to one species are black.

remained similar to (or even a little stronger than) that of wild-type hFc. From these results, we conclude that RbF4 likely contacts H310 and N315, but not H435.

Seven RbF4-hFc docking models were consistent with the binding results; RbF4 makes contacts with H310 and N315, but not H435 or the six other positions mutated in Vars 1 and 2 (**Fig 1C**). We hypothesized that the all-atom force field energy could largely capture the binding free energy landscape. The seven docking models were ranked by total energy according to the AMBER99sb force field [32] after full-atom minimization using the Tinker molecular dynamics package [33]. We determined the complex structure by X-ray crystallography (deposited as PDB ID: 6KA7), and comparison of the crystal structure with the docking models confirms that the binding mode of the docking model with the lowest energy (**Fig 1C** and **1D**, blue model) and that of the crystal structure are indeed extremely similar ($f_{nat}$: 0.43 and I-RMSD: 2.89 Å, see **Fig 2** and **S2 Table**). The full atom structure minimization changed the overall Fc structure, and consequently one complex model had a slightly better I-RMSD than the lowest-energy model, but the lowest-energy model better maintained interactions across the interface and thus had a much better $f_{nat}$ (**S3A Fig**). These results demonstrate the utility of our integrated computational and experimental approach to identifying a native-like complex model for an LRR protein: first a computational method designs sets of mutational variants to probe docking models; then experimental binding assays effectively filter the docking model candidates; finally full-atom minimization ranks the filtered docking models. It is noteworthy that epitope localization is an essential step for precise binding mode identification, and ranking docking models using the force field energy alone may be insufficient for finding a native-like model (**Fig 2B** and **2C**). Furthermore, testing the individual mutations comprising a selected variant was important in this case; if all of the three positions in Var 3 were assumed to be important, there would be only two possible docking models (**S3B Fig**), and while the interface region of the lower energy model is largely correct, its binding orientation is completely reversed.

## Redesign of binding specificity based on the modeled complex

Based on the model complex structure, we redesigned RbF4 to alter its binding specificity. RbF4 was previously determined to be highly specific for human IgG$_1$, showing weak and negligible cross-reactivities against mouse IgG$_1$ and rabbit IgG [30, 31]. The confirmed complex model here reveals that the loop (**S1 Fig**) may be largely responsible for the binding specificity of RbF4 toward hFc (**Fig 1D**). To obtain further insight into this possible source of specificity, we investigated the Fc sequences of IgGs from three species: human (hFc), mouse (mFc), and rabbit (rFc). The modeled RbF4:hFc complex shows that the RbF4 loop forms a tight contact with the positions where amino acids differ among hFc, mFc, and rFc (**Fig 1D**), strongly supporting a crucial role for this loop in the observed binding specificity of RbF4. We thus reasoned that engineering the loop could yield a variant of RbF4 showing cross-reactivities for Fc from other species. To prove our hypothesis, we replaced the loop sequence of RbF4 starting at position 239, RNSAGSVA, with the truncated but flexible amino acid pair GG. We measured with ITC assays the binding affinities of the resulting loop-truncated RbF4 variant (RbF4-LT) against the IgGs from the three species: hIgG$_1$ (Trastuzumab), mouse IgG$_1$ (mIgG$_1$), and rabbit IgG (rIgG).

As shown in our previous work [31], the original RbF4 binds strongly to hIgG$_1$ with a binding affinity of 128.7 nM, whereas binding weakly to mIgG$_1$ with an 8-fold lower affinity of 1 μM, and has a negligible binding affinity for rIgG (**Fig 3** and **S4 Fig**). The loop-truncated variant (RbF4-LT) displayed improved binding affinity for rIgG (1.2 μM), indicating that the loop is indeed involved in the binding specificity of RbF4 for hIgG$_1$. Its binding affinity for mIgG$_1$

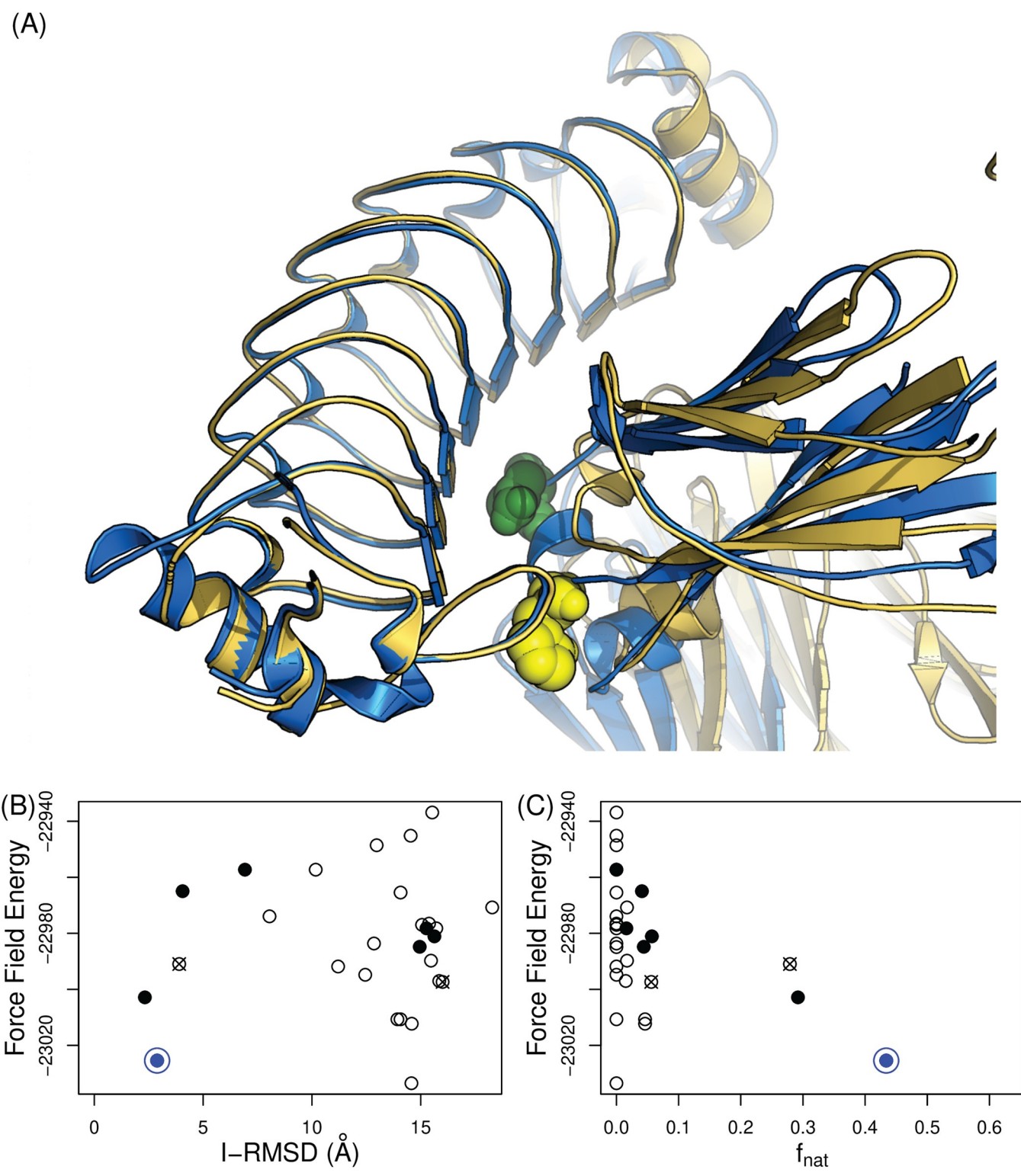

**Fig 2. Computationally-driven identification of the RbF4-hFc complex.** (**A**) The lowest-energy model is in blue and the crystal structure (6KA7) is in gold. H310A and N315K are highlighted in spheres (**B,C**) Comparison model energy vs. (**B**) I-RMSD and (**C**) $f_{nat}$ Docking models that are in contact with the epitope residues (correctly localized docking models) are shown with solid circles. The crossed-circles are models in contact with all of the three residues in Var 3. The blue circle is the model with the lowest force field energy (AMBER99sb) score (illustrated in panel A).

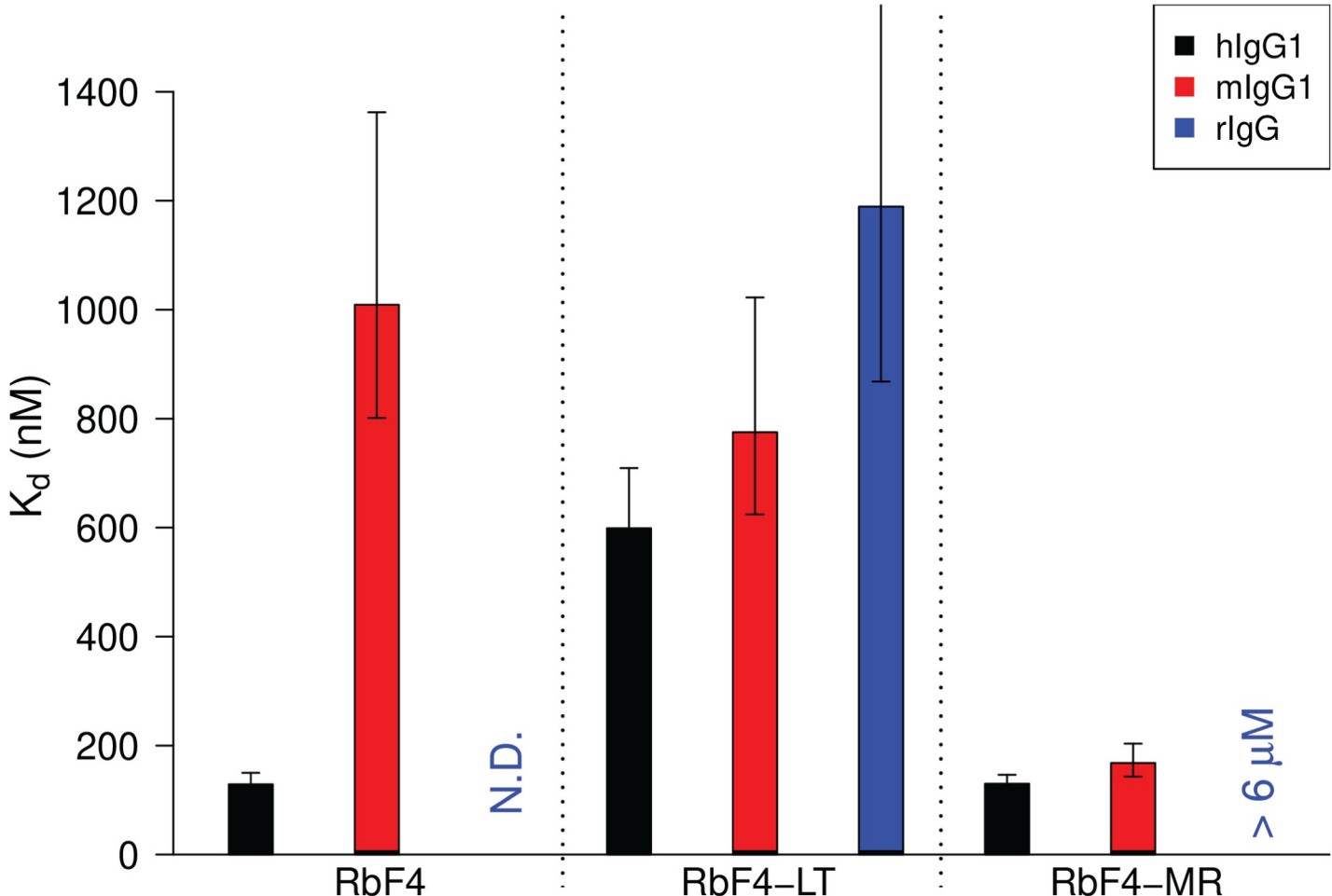

**Fig 3. Redesigning binding specificity and affinity based on modeled complex structure.** The model suggests that the binding specificity of RbF4 comes from the loop (**Fig 1D**). The loop truncated RbF4 (RbF4-LT) was testing for binding against the IgGs from the three species. Further computational design on the model identified mutations (S241M and S244R: RbF4-MR) that could significantly improve the binding affinity for mIgG$_1$ (1 μM to 168.1 nM) while maintaining that for hIgG$_1$ (**S3 Table** for details). ITC-based affinity measurements were performed in triplicate.

was also improved (775 nM), whereas that for hIgG$_1$ decreased to a level similar to that for mIgG$_1$ (598 nM). It bears noting that the variant was designed based only on the modeled binding mode, prior to the determination of the X-ray co-crystal structure. The results thus demonstrate that our integrated approach can provide sufficiently accurate complex models for the redesign of binding specificity.

## Improvement of binding affinity based on the modeled complex

Our final goal is to use computational design to improve the binding affinity of RbF4 against different IgGs. As we saw in determining the best complex model, while the full-atom force field energy alone did not identify the near-native complex, it did largely capture the binding energy landscape. We thus again used the AMBER99sb total energy to select loop mutations predicted to simultaneously improve binding affinities of RbF4 against multiple targets. To reduce the search space, FoldX [34] was employed to fast-scan possible mutations in the loop (**S5 Fig**), since we observed that FoldX is particularly accurate in predicting disruptive mutations (PPV > 0.9 for antibody-antigen pairs) [35]. The predicted binding energy values at

G243 indicate that the inclusion of the loop may not enhance a binding affinity for rIgG. Thus we aimed to design an RbF4 variant which can bind to both $hIgG_1$ and $mIgG_1$ with high binding affinities. The FoldX scan suggests that S241M may substantially enhance the binding affinities for $hIgG_1$ and $mIgG_1$. The mutation was also observed during the phage display affinity maturation for the $mIgG_1$-specific repebody [31]. We then fixed S241M and introduced all other amino acids *in silico* at S244. The AMBER99sb force field energy was used to minimize the variants. The binding energy prediction indicated that S241M with S244R (RbF4-MR) may significantly improve the binding affinities for both IgGs (**S6 Fig**). We tested this variant, and the ITC binding assay indeed showed that RbF4-MR strongly binds to the two IgGs as predicted, and the binding affinity for $mIgG_1$ was markedly increased (1 μM to 168.1 nM, **Fig 3**).

## General applicability of the binding mode identification method

To assess the general applicability of our integrated approach to identifying a binding mode, we investigated three known Rb targets whose co-crystal structures are available (**Table 1**): Interleukin-6 (IL-6: 4J4L) [29], epidermal growth factor receptor (EGFR: 4UIP) [36] and complement component 5a (C5a: 5B4P) [37]. We first investigated the importance of using ClusPro's "antibody mode" for Rb docking. Docking without the antibody mode option resulted in a larger number of docking models, but the overall accuracy of the docking models proved to be worse than that of those generated with the option enabled (**Table 1**). The antibody mode puts a lower weight on the DARS [38] energy term than other docking modes [32]. In order to improve binding affinities of antibodies and other protein binders, mutations are extensively made on only one of the interacting partners (e.g., complementarity determining regions in antibodies) [39]. Thus, the statistics of observed amino acid frequencies for the binding interface regions are different. DARS assumes a symmetry interaction, which is beneficial for general protein-protein docking, whereas it is worse for antibody-antigen pairs [39]. We likewise hypothesize that interaction assymetry, captured in ClusPro's antibody mode, leads to improved prediction accuracy for repebody binding and other affinity matured protein binders.

Consistent with the results above for RbF4-hFc, only a small number of variants were sufficient to localize the epitopes (two for C5a and EGFR, and four for IL-6). The filtering process resulted in a small set of docking models including native-like ones (5 to 12 models; solid circles in **Fig 4**). In order to identify the most native-like docking model, the ClusPro score was initially considered to rank the filtered docking models, but it was observed to be unreliable (**S7 Fig**). In constrast, but consistent with the RbF4-hFc results, **Fig 4** shows that ranking based on the AMBER99sb force field energy successfully discriminated high-quality docking models for IL-6 and EGFR. It bears noting that the prior epitope localization was again critical; ranking by the force field energy alone was not sufficient to find native-like docking models. For example, in the case of IL-6, there are two incorrect docking models with lower energies than the most native-like model, but both of them are not in contact with true epitope residues and thus were filtered out.

The C5a case provides an additional insight into precise binding mode prediction (**Fig 4**, bottom row). While the binding interfaces were mostly correct (15 out of 18 correct interface residues: 83%), the predicted binding mode was completely inverted (N terminal to C, and vice versa). During the affinity improvement of the C5a-specific Rb, it was observed that some LRRV modules gave rise to a negligible increase in the binding affinity [37], suggesting that only LRRV1 and 2 were responsible for interacting with C5a. We thus hypothesized that the accuracy of structural modeling and an incorporation of the paratope information may also enhance docking quality. The four possible combinations of hypotheses (incorporation of the

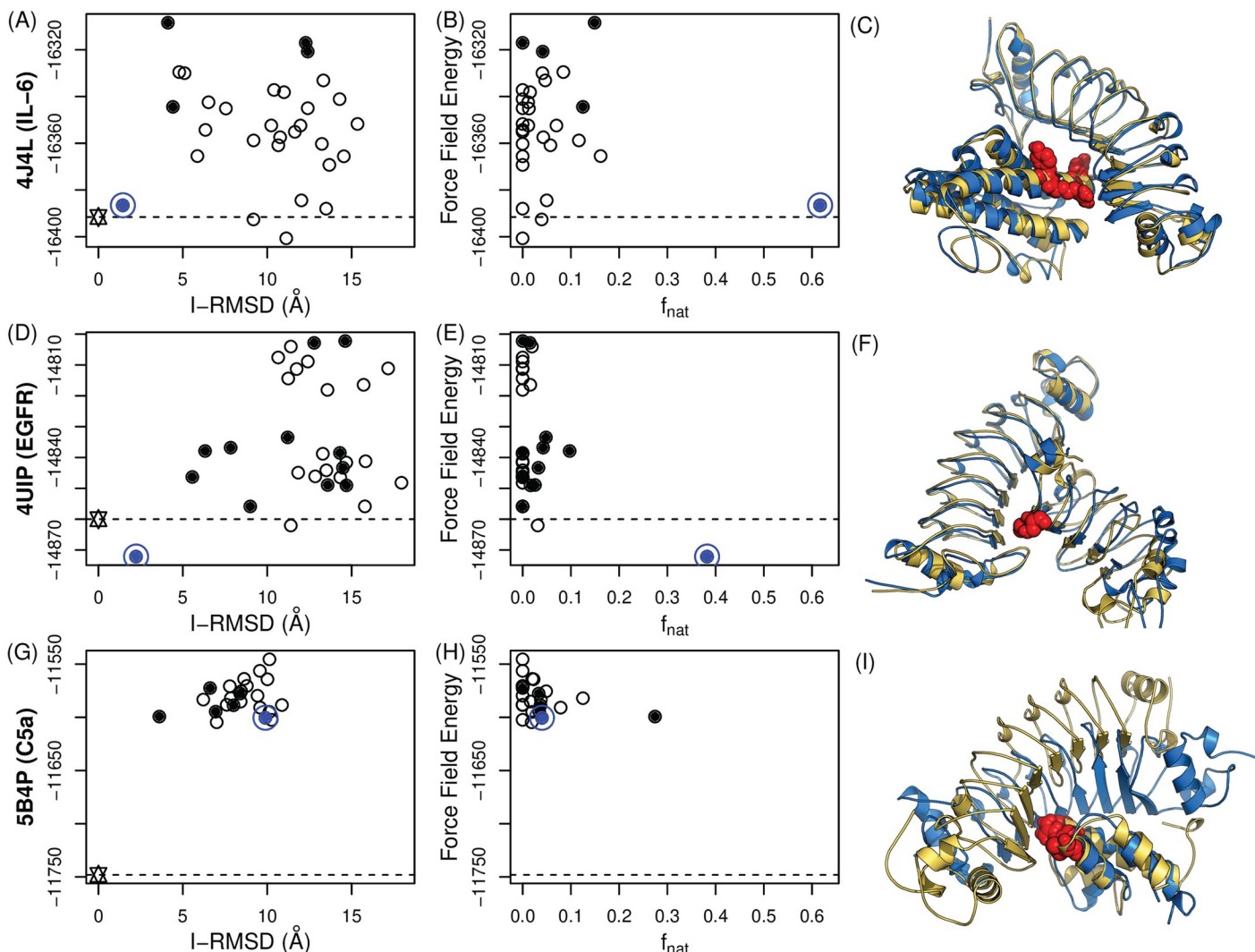

**Fig 4. Retrospective tests of binding mode identification for additional targets.** (**A**, **B**, **C**) IL-6; (**D**, **E**, **F**) EGFR; (**G**, **H**, **I**) C5a. (**A**, **D**, **G**) I-RMSD vs. energy; (**B**, **E**, **C**) $f_{nat}$ vs. energy; (**C**, **F**, **I**) Points represent docking models, with those that are in contact with epitope residues (correctly localized docking models) shown as solid circles. The blue circle is the model with the minimum force field energy (AMBER99sb), and is native-like for IL-6 and EGFR. Wild-type (star) energy levels are depicted as dotted lines. For IL-6, some models have lower energy values than the most native-like docking model, indicating that ranking only by the force-field energy is not sufficient for binding mode prediction. (**C**, **F**, **I**) Crystal structures are in gold and the docking models with the lowest energies are in blue.

phage display information versus no incorporation, and a homology-model target versus a crystal structure target) revealed that the use of a high-quality structure (here the crystal structure) was not sufficient for accurate binding mode identification (**Fig 5**). No paratope information with the crystal structure resulted in worse correlations than the model with precisely annotated paratopes (Spearman ρ for $f_{nat}$ and I-RMSD of crystal structures: -0.1 and 0.1, and those of models with precise paratope definition: -0.7 and 0.19, respectively). The ideal case (crystal structures with precise paratopes) led to extremely accurate results (I-RMSD: 0.62Å and $f_{nat}$: 0.76) with high correlations to the crystal structure (Spearman ρ for $f_{nat}$ and I-RMSD: -0.79 and 0.71).

This retrospective study demonstrated critical criteria for accurate binding mode identification. The full-atom force field energy can effectively discriminate the most native-like docking model when combined with an initial epitope localization using experimental data to filter the

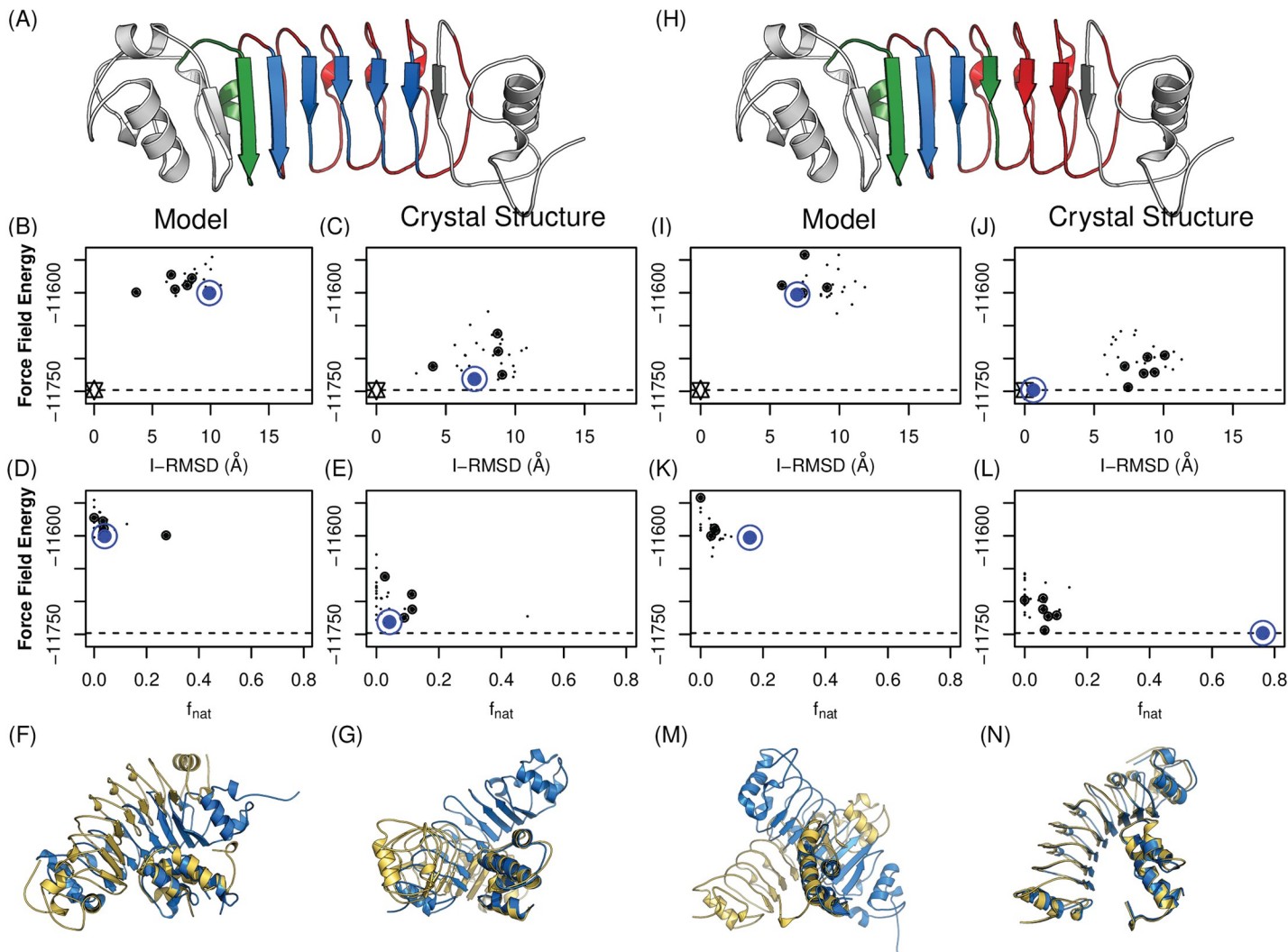

**Fig 5. Impacts of high-quality structure and paratope definition.** The incorporation of phage display results and the use of high quality
structures (here crystal structures) lead to an extremely accurate identification of the binding mode. LRRV in blue is assigned for attraction, in red for repulsion and green for neutral (**A** and
**H**). Results in the right two columns (**I-N**) are from docking models with precisely assigned paratopes (Results with no paratope information are in the left (**B-G**)).
Docking models that are in contact with epitope residues (localized docking models) are in solid circle. Models with the minimum force field energy values are in blue
circles. Wild-type (star) energy levels are depicted as dotted lines. Crystal structures are in gold and the docking models with the lowest energies are in blue.

models; on its own it may not be sufficient. As also observed in the previous study with anti-
bodies [11], high-quality antigen structures are not required for epitope localization; homology
models generally suffice. However, they are necessary to generate and predict native-like bind-
ing modes. Finally, while, like antibodies, repebodies have well-defined target-binding sites,
not all of them are actually involved in target binding. The inclusion of paratope information
(e.g., residues contributing to phage display selection) was shown to improve the quality of
docking models and binding mode prediction.

## Discussion

Precise binding mode identification of protein binders is crucial for the development of thera-
peutic proteins, but they have heavily relied on labor-intensive experimental approaches.
Computational methods that do not require structure determination offer a way to accelerate

development processes and understanding of mechanisms of action, advancing the potential utility of relevant proteins in translational and basic research. We chose LRR proteins as a model system to evaluate the utility of an intergrated computational and experimental approach to identifying binding modes and reengineering binding affinity and specificity, since theses proteins not only represent a promising therapeutic scaffold, but they also have the important advantage of undergoing small structural changes upon target binding. We extensively studied one LRR protein binder targeting the constant domain of human IgG$_1$ (hFc). Computaional and experimental epitope localization followed by full-atom energy minimization with the AMBER99sb force field enabled the successful selection of a docking model which was confirmed to be most native-like according to the independently solved X-ray crystal structure. The further utility and potential of our computationally-guided binding mode identification were demonstrated by successfully implementing the resulting model to design variants with increased binding affinity and altered specificity.

Unlike epitope localization or binding site identification, which do not require extremely native-like binding modes in the sampling step [11], the inclusion of native-like complex models is absolutely necessary for further affinity improvement from models. In general, the sampling quality of antibody-antigen docking often depends on the targets; however, in our case, native-like models were nearly always included among the samples, likely due to the rigid nature of LRR domains. Therefore, we anticipate that this approach may in principle be applicable not only to LRR domains, but also to any other proteins with rigid binding sites. As also demonstrated by the success of interface-guided docking methods [40–42], the exact definition of binding interfaces including paratope mutational data from phage display selection is also important.

As repeatedly shown here, selection of a docking model based only on the molecular modeling energy may be misleading, perhaps due to the inaccuracy of the energy functions [43]. Mutagenesis-based verification and filtering of docking models generally focused on the correct epitope region, and they were shown to be necessary here in order to complement imperfect energy scores. The epitope localization method used in this study, EpiScope [11], provided an optimal set of mutational variants; here requiring only six sets of *in vitro* experiments to effectively localize the epitope. With the epitope thereby localized, full-atom minimization enabled ranking of filtered docking models, resulting in identification of a native-like docking model.

The successful modeling of binding mode directly enabled the design of binding specificities and affinities. From the model, we were able to identify key contributors to the binding specificity of RbF4 for the IgGs and engineer a simple manipulation to entirely change its binding specificity. Furthermore, since the force field energy is indicative of binding energy, we were able to select mutations based on the energy and dramatically improve binding affinities as predicted. However, as discussed above, this holds only with a good model of binding, as otherwise design only based on the energy may result in wrong predictions. For example, the force field energy prediction suggeests that RbF4-MR would also have a good binding affinity to hIgG$_1$ though the actual binding affinity was not improved (**S7 Fig**).

## Methods

### Computational methods

To design mutations for epitope localization, we used EpiScope with default settings [11]. three mutations per binding patch, and one pareto-optimal curve and two suboptimal ones. The docking models were generated using the ClusPro webserver [44], initially using "Antibody mode" [39], and later a separate set was generated without that option. The non-CDR

masking option was disabled, but the binding sites were masked for attraction and the convex side for repulsion (**S1A Fig**). As with the models of the targets, the complex models were minimized using the Tinker molecular dynamics package [33] with the AMBER99sb parameter set [32] and the GB/SA implicit solvent model [45]. The total energy was used to rank complex models, and FoldX (ver. 4) was employed to fast filter binding disruptive mutations [34]. A complex structure is repaired and optimized using 'RepairPDB'. Given a mutation or set of mutations, the effect on binding is then calculated using the 'BuildModel' command.

There are currently three Rb-target complex structures in the PDB: binders to IL-6 (PDB code 4J4L), EGFR (4UIP), and C5a (5B4P). Unbound forms of the target structures were used for docking (**Table 1**). There is a missing loop in the structure of IL-6 (1ALU chain A: 52 SSKEALAEN). The loop was filled using MODELLER [46] and all the backbone atoms of the loop were minimized using Tinker as described above. While repebodies have a very rigid predefined structure, a single mutation at the 11$^{th}$ LRRV position (**S1B Fig**) to proline significantly changes the conformation. Two of the Rbs (4J4L and 5B4P) have such proline residues (at LRRV1 and LRRV2 respectively). These two Rbs were modeled using each other as templates. One LRRV unit of the EGFR Rb (4UIP) was omitted. The trimmed Rb structure was reconstructed by splitting the free Rb at LRRV3 and superimposing the LRRV3 on LRRV2 of the complete Rb using PyMol.

## Preparation of Fc variants

The sequence of the human Fc binder repebody (RbF4) was obtained from a previously published study [30]. The Rb structure was modeled using MODELLER with a free form (3RFS:A) as a template structure. A free form of the hFc domain (3AVE) was used. Trastuzumab (trade name, Herceptin) Fc sequence available from the literature (wild-type) and all subsequent variants were reverse translated, codon optimized for expression in mammalian cells, and synthesized by Integrated DNA technologies (IDT), Inc. (Redwood City, CA). CMVR VRC01 expression vector (NIH AIDS reagent program, Germantown, MD) harboring the wild-type Fc or the Fc variant sequences was transfected into suspension HEK 293 cells using polyethylenimine (PEI) (Polysciences, Warrington, PA). Briefly, 500 μg of the wild-type Fc or Fc variant DNA was combined with 1 ml of PEI and incubated at room temperature for 10 minutes. The mixture was then added to HEK cells in the suspension and incubated in a humidified chamber at 37˚C with 8% $CO_2$ for at least 5 to 6 days. The secreted wild-type Fc or Fc variants were clarified through centrifugation at 8000 rpm at 4˚C for 15 minutes on a Beckman Avanti-J25 centrifuge (Brea, CA). The resulting supernatant was filtered through a 0.45 μm filter to remove any residual cell debris and other large particles before loading onto a FPLC column.

Affinity purification was conducted on a pre-packed 5 ml Protein A column (for wild-type Fc, and Fc variants 1 and 2) or pre-packed 5 ml Protein G column (for Fc variant 3, and single mutations of the variant) from GE Healthcare (Pittsburgh, PA) as suggested by the manufacturer. The final sample was eluted with 100 mM Glycine at pH 3 in 2 ml Eppendorf tubes prefilled with 50 μl of 1 M Tris and 5 mM EDTA. The purification process was automated on an AKTA FPLC system (GE Healthcare, Pittsburgh, PA). The purified protein was subjected to a second buffer exchange step using a hitrap desalting column (GE Healthcare, Pittsburgh, PA). The final product was eluted in phosphate buffer saline and stored at -20˚C until further use. The purified proteins were analyzed under reduced SDS-PAGE conditions and stained with coomassie blue.

## Expression and purification of repebodies

The gene-encoding repebodies were inserted into NdeI and XhoI restriction sites of a pET21a vector (Invitrogen, Carlsbad, CA). Plasmids were cloned into competent *E.coli* DH5α cells

using a heat shock method (at 42˚C for 90 seconds). The recombinant plasmids were transformed into *E.coli* Origami-B cells (Merck, Kenilworth, NJ). Single colonies were inoculated into 5 mL of a Luria-Bertani (LB) medium containing 50 μg/mL carbenicillin and grown overnight at 37˚C in a shaking incubator (200 rpm). A total of 250 mL of LB containing 50 μg/mL carbenicillin was inoculated with an $OD_{600}$ 0.05 volume of the overnight-saturated culture and grown at 37˚C with shaking at 200 rpm until the $OD_{600}$ reached 0.5–0.8. The cells were induced using 0.5 mM IPTG and incubated at 18˚C with shaking at 200 rpm for 16 hours. The cells were harvested through centrifugation at 8000 rpm for 20 mins and suspended in a lysis buffer (50 mM $NaH_2PO_4$, 300 mM NaCl, and 10 mM imidazole, at pH 8.0). After cell lysis by sonication, the cell debris was removed through centrifugation at 16,000 rpm for 1 hour at 4˚C. Cell lysates were loaded into a Ni-NTA column (Qiagen, Hilden, Germany) and washed using a wash buffer solution (50 mM $NaH_2PO_4$, 300 mM NaCl, 20 mM imidazole, at pH 8.0). The repebodies were eluted using an elution buffer (50 mM $NaH_2PO_4$, 300 mM NaCl, 250 mM imidazole, at pH 8.0), and further purified using gel permeation chromatography (Superdex 75, GE Healthcare). The buffer of the purified repebodies changed into PBS, and the concentrate was developed in an AmiconUltra centrifugal filter (10 kDa cutoff, Millipore).

## Determination of crystal structure

The Fc domain of human IgG (hFc) was purified after digestion of the purchased human IgG (Sigma, St. Louis, MO) with papain, as described elsewhere [47]. The RbF4-hFc complex was purified through gel-filtration with a buffer containing 5 mM Tris·HCl and 0.1 M NaCl (pH 7.4) after reconstitution of the complex by incubating RbF4 and hFc at a 2:1 molar ratio on ice. Crystals of the complex were grown using a hanging drop vapor diffusion method against a crystallization buffer containing 0.1 M sodium acetate, 12% (w/v) polyethylene glycol 6000, and 0.1 M magnesium chloride (pH 4.6) at 20˚C. The crystals formed in the space group $P2_12_12_1$ with $a$ = 59.9 Å, $b$ = 107.4 Å, and $c$ = 171.4 Å, and contained one complex molecule in an asymmetric unit. The diffraction data were collected at 100 K, with crystals flash-frozen in a crystallization buffer containing 30% glycerol. A single-wavelength (1 Å) dataset was collected using a native crystal on beam line 5C (Pohang Accelerator laboratory, Korea). Integration, scaling, and merging of the diffraction data were conducted using the HKL2000 program suite [48]. The initial phases were determined through molecular replacement using the Phenix AutoMR program [49] and human Fc of IgG (PDB accession 1H3X) and repebody (PDB accession 3RFJ) as models. Successive rounds of model building using Coot [50], refinement using the Phenix program [51], and phase combination allowed the building of the complete structure (**S1 Table**).

## Isothermal Titration Calorimetry (ITC)

A binding affinity experiment was conducted using MicroCal-iTC200 (Malvern Instruments, Malvern, UK). Fc variants, mouse $IgG_1$, and rabbit IgG (Sigma) were diluted in a PBS buffer to a final concentration of 0.02 mM. RbF4 or RbF4-LT (RbF4 with the loop truncation) was diluted using the same buffer to a final concentration of 0.2 mM. The ITC experiments were performed for a total 20 injections and stirred at 1000 rpm. The initial injection of 1 μL was excluded for data analysis. Titration curves were fitted with a one-site binding model. The value of $K_d$ was determined using Origin (OriginLab).

## Supporting information

**S1 Fig. The repebody structure.** (**A**) A repebody (Rb) largely consists of three parts: N-termianl cap (LRRNT), variable regions (LRRV) and C-terminal cap (LRRCT). Binding occurs at

the concave region of LRRV (in darker blue). (**B**) Structure of a single LRRV motif, with side chains of conserved residues rendered as stick figures. Each LRR is composed of six conserved leucine residues, a central conserved asparagine residue, and conserved phenylalanine residue on the C-terminal side.
(TIF)

**S2 Fig. Detailed information about the mutations for hFc-F4 epitope localization and titration curves.** Based on the $K_d$ values, H310 and N315 overlap epitopes.
(TIF)

**S3 Fig. Docking models of RbF4.** The crystal structure is in gold. (A) The full atom energy minimization step may change the overall structure, but the binding interactions are likely to be maintained. The I-RMSD value of the lowest energy model (Model 1, blue) is slightly higher than the model with the second lowest energy (Model 2, pink). However, its $f_{nat}$ is twice higher. (B) There are two docking models which are in contact with the three mutations in Var 3 (Model 9: cyan and Model 10: pink). While their binding interface regions are largely correct, the binding orientation of the lower energy model (pink) is completely inverted.
(TIF)

**S4 Fig. Details of Binding affinities of RbF4 variants (RbF4, RbF4-LT and RbF4-MR) for hIgG$_1$, mIgG$_1$ and rIgG.** RbF4 binds strongly to hIgG$_1$ and weakly to mIgG$_1$. However, no binding affinity is measured for RbF4-rIgG. The truncation of the loop (RbF4-LT) enabled the variant to bind to all IgGs with similar binding affinities. RbF4-MR gains strong binding affinities for hIgG$_1$ and mIgG$_1$. See **S2 and S3 Tables** for details.
(TIF)

**S5 Fig. FoldX scan of the RbF4 loop.** The residue scan using FoldX suggests that the inclusion of the loop may not enhance the binding affinity of RbF4 for rIgG.
(TIF)

**S6 Fig. Predicted ΔΔG values of the RbF4 variants.** The variant with S241M and S244R mutations (RbF4-MR) is predicted to strongly bind to both hIgG$_1$ and mIgG$_1$. S244C and S244P were not considered.
(TIF)

**S7 Fig. Binding mode prediction with the ClusPro score.** The ClusPro score was tested on the retrospective test set (**A-C**: IL-6, **D-F**: EGFR, and **G-I**: C5a binders). Docking models that are in contact with epitope overlapping residues (localized docking models) are in solid circle. The blue circle is the model with the lowest ClusPro score. Crystal structures are in yellow and the docking models with the lowest energies are in blue on the right hand side. Score assessment using the ClusPro score is not predictive.
(TIF)

**S1 Table. Data collection and refinement statistics.**
(DOCX)

**S2 Table. Quality measures and contact information of RbF4 docking models.** Each black block indicates that the Fc position is in contact with the docked repebody.
(DOCX)

**S3 Table. *In silico* binding specificity control and affinity improvements.** RbF4 strongly binds to hIgG$_1$ (weakly to mIgG$_1$), but no binding to rIgG is observed. Removal of the loop (RbF4-LT) causes a slight reduction in the binding affinity toward hIgG$_1$, but yields a marginal

affinity improvement for mIgG$_1$. RbF4-LT produces a significant improvement in the affinity for rIgG. RbF4 with two mutations at 241 and 244 (S241M with S244R, RbF4-MR) binds to both hIgG$_1$ and mIgG$_1$ with high binding affinities.
(DOCX)

## Author Contributions

**Conceptualization:** Yoonjoo Choi, Hak-Sung Kim.

**Data curation:** Yoonjoo Choi, Sukyo Jeong, Jung-Min Choi, Christian Ndong.

**Formal analysis:** Yoonjoo Choi, Sukyo Jeong, Jung-Min Choi.

**Funding acquisition:** Yoonjoo Choi, Karl E. Griswold, Chris Bailey-Kellogg, Hak-Sung Kim.

**Investigation:** Yoonjoo Choi, Sukyo Jeong, Jung-Min Choi, Christian Ndong.

**Methodology:** Yoonjoo Choi.

**Project administration:** Karl E. Griswold, Hak-Sung Kim.

**Resources:** Christian Ndong, Karl E. Griswold, Chris Bailey-Kellogg, Hak-Sung Kim.

**Software:** Yoonjoo Choi.

**Supervision:** Karl E. Griswold, Chris Bailey-Kellogg, Hak-Sung Kim.

**Validation:** Yoonjoo Choi, Sukyo Jeong, Jung-Min Choi, Christian Ndong.

**Visualization:** Yoonjoo Choi, Jung-Min Choi.

**Writing – original draft:** Yoonjoo Choi, Sukyo Jeong, Hak-Sung Kim.

**Writing – review & editing:** Yoonjoo Choi, Sukyo Jeong, Jung-Min Choi, Karl E. Griswold, Chris Bailey-Kellogg, Hak-Sung Kim.

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
