## [Decision Letter · Decision Letter 0]

18 May 2020

Dear Dr. Kim,

Thank you very much for submitting your manuscript "Computer-aided Binding Mode Prediction and Affinity Maturation of an LRR Protein Binder without Structural Determination" for consideration at PLOS Computational Biology.

As with all papers reviewed by the journal, your manuscript was reviewed by members of the editorial board and by several independent reviewers. In light of the reviews (below this email), we would like to invite the resubmission of a significantly-revised version that takes into account the reviewers' comments.

Take particular care in addressing comments related to your previously published paper ref. 10 and explain more in details the differences and advancements produced in the current submission. Also, please have the English double-checked. 

We cannot make any decision about publication until we have seen the revised manuscript and your response to the reviewers' comments. Your revised manuscript is also likely to be sent to reviewers for further evaluation.

Sincerely,

Marco Punta

Associate Editor

PLOS Computational Biology

Arne Elofsson

Deputy Editor

PLOS Computational Biology

Reviewer's Responses to Questions

**Comments to the Authors:**

Reviewer #1: Recommendation: Publish with minor corrections

Comments:

This paper describes the application of protein:protein docking in conjunction with epitope prediction and experimental mutation data to predict the complex of an LRR protein and IgG1. This article is extremely well written and is a beautiful example of the power of computational techniques when paired with experimental data to guide the design of protein binders. I therefore recommend publication of the article with minor corrections.

My first concern is that the authors seem to make generalizations in the introduction but cite single articles which make the point. While the generalizations are correct there are better references the authors could cite.

Page 4 line 64 the authors cite references 10, 11 for the current status of prediction protein:protein binding poses. The authors should cite a comparative study such as:

1. Huang S-Y. Exploring the potential of global protein–protein docking: an overview and critical assessment of current programs for automatic ab initio docking. Drug Discov Today. Elsevier Ltd; 2015;20: 969–977. doi:10.1016/j.drudis.2015.03.007

2. Hogues H, Gaudreault F, Corbeil CR, Deprez C, Sulea T, Purisima EO. ProPOSE: Direct exhaustive protein-protein docking with side chain flexibility. J Chem Theory Comput. 2018;14: 4938–4947. doi:10.1021/acs.jctc.8b00225

Page 4 line 64 the authors cite reference 12 with respect to the challenge of prediction protein:protein affinity. The authors should cite a comparative study or a more relevant review such as:

1. Sirin S, Apgar JR, Bennett EM, Keating AE. AB-Bind: Antibody binding mutational database for computational affinity predictions. Protein Sci. 2016;25: 393–409. doi:10.1002/pro.2829

2. Kastritis PL, Moal IH, Hwang H, Weng Z, Bates PA, Bonvin AMJJ, et al. A structure-based benchmark for protein-protein binding affinity. Protein Sci. 2011;20: 482–491. doi:10.1002/pro.580

3. Siebenmorgen T, Zacharias M. Computational prediction of protein–protein binding affinities. Wiley Interdiscip Rev Comput Mol Sci. 2019; 1–18. doi:10.1002/wcms.1448

While not a concern, on Page 6 line 113-115 the authors state that they “… assumed that the major driving force of RbF4 for target binding should be no different from antibodies or high affinity protein binders (Antibody Mode)”. While the authors clearly show the benefit of using the Antibody mode scoring function in ClusPro in table 1, I would like the authors to expand on this statement and explain which features they feel are similar between their binding event and that of antibody CDR interacting with an antigen. In the reference cited by the authors (reference 28) the ClusPro authors state that antibody recognition is typically flatter and less hydrophobic than enzyme pockets, is this similar for your binding event. Additional discussion and insight would be beneficial to the reader.

Page 7 line 136 does the author mean AMBER99sb forcefield energy? Is this the total energy including the internal energy (bond, angle, torsions)?

On Page 9 Line 172-173 the authors state the following “The results indicate that the predicted binding mode is sufficiently accurate for further engineering of the binding specificity.” While the authors results are great, we should be careful about people misreading this sentence. The authors should clearly state that the predicted binding mode is the result of not only using a docking tool and scoring function but results were filtered using experimental information.

On page 9 line 179 the authors mention selection of LRR designs using AMBER forcefield energy yet this is not described in the experimental protocol on page 14. Did you use AMBER99sb again? Did you use total energy or MM-GB/SA? Clarification is needed.

On page 9 line 180 the authors mention a FoldX scan, but no mention of this in the computational methods section. Could the authors please include this. Also, to follow up on the above point did the authors solely use FoldX to create the mutants then use AMBER to score them or did they use the FoldX binding affinity scoring function? Again, further clarification is needed.

On Page 14 line 280-283 the authors state “From the two studies of the binding mode prediction and affinity improvement, the force field energy prediction becomes informative only if actual binding is known, i.e. supportive data from experiments are critical in practice.” I believe the authors are referring to their work here and therefore they should explicitly state that. Also, the statement is partially incorrect, since the authors clearly show that the forcefield energy can discriminate docking poses, but not how it correlates with binding affinity. This should be removed.

Overall this is an excellent paper that I can highly recommend for publication after minor corrections.

Reviewer #2: The manuscript by Choi et al. entitled "Computer-aided Binding Mode Prediction and Affinity Maturation of an LRR Protein Binder without Structural Determination" describes a project in computational protein engineering in which epitope prediction, docking and computational energy minimization were combined with intermediate experimental validation steps to determine the binding mode of the RbF4-hFc interaction to facilitate affinity maturation of this interaction. This work closely follows the approach taken in a previous publication (ref [10]), extending to the context of a leucine-rich repeat protein binder.

The ultimate goals of the work were three-fold:

i. To test the ability of computational energy minimization to accurately predict the binding mode of the interaction between hFc and a previously developed LRR protein binder or repebody RbF4, and to validate the accuracy of the prediction.

ii. To evaluate the utility of these predictions for affinity maturation of this interaction.

iii. To demonstrate that this approach can be applied to other LRR protein binders.

Overall, this is an interesting study that represents a contribution to the growing literature surrounding the application of computational approaches to the engineering of protein binders.

Major comments:

The authors use existing tools (ClusPro in conjunction with EpiScope) to predict the epitope of the RbF4-hFc interaction, using a crystal structure of the unbound hFc, and a homology model of the repebody RbF4. The strategy used in this study is initially entirely consistent with that in ref. 10, and the novelty here is that it is applied in the context of the RbF4-hFc interaction. Verifying that the strategy presented in ref. 10 can be applied in this additional context is useful.

1) The authors need to make clear in the introduction to the study the relationship of the work carried out in this manuscript to the previously published protocol from ref [10]. It is not immediately clear to the reader that the approach described in this manuscript is not wholly original, and I found that to be somewhat misleading. Placing the work in the broader context of the field does not detract from it.

2) How did the authors confirm that the measured reduction in binding affinity was not caused by changes in the stability or structure of Var3, H75A or N80K, but rather resulted from specific disruption of the Ab-Ag binding interface?

3) What positions were contained in Var 1 and Var 2? Please provide position numbering that is concordant the the PDB structure 3AVE referred to in the text.

4) The decision to evaluate each of the three mutations in Var 3 as single mutants appears to deviate from the approach proposed in ref 10, requiring additional experimental work. The authors comment:

Lines 125-126 'Although the hFc interface region in contact with RbF4 was roughly identified through the triplet, it is probably that not all of them are involved in the binding.'

What is the effect on the subsequent analysis if this step to interrogate each of the three mutations in Var 3 is not taken? What do the authors recommend to future users of this technology as the standard approach?

5) Figure 2B and C show that of the docking models in contact with the epitope overlapping residues, the model with the lowest energy has close to the lowest I-RMSD and the highest f_nat. How different is the binding mode of the model with higher energy and lower I-RMSD?

6) Similarly, how different were the models with lower energy in Figs 2B and C that did not fulfill the conditions of being in contact with H75 and N80 but not with H200 or the six other positions in Var 1 and 2? Please provide a table in the supplement with details of the positions in contact in each model, and the force field energy and structural parameters (I-RMSD, f_nat) for each of the 29 models.

Minor comments:

1. While overall the writing and presentation is fully adequate and relatively easy to understand, the paper would nonetheless benefit greatly from careful reading to fix a large number of awkward and/or unclear statements and phrases. For example:

(i) Lines 111-113 are quite unclear, in particular the phrase 'assigning attractions at the concave residues of LRRV modules from 2 to 4, as known during the phage display selection' does not make sense.

2. What is shown in red in Figure 1d? I'm guessing it is the repebody loop?

**Have all data underlying the figures and results presented in the manuscript been provided?**

Reviewer #1: Yes

Reviewer #2: Yes

PLOS authors have the option to publish the peer review history of their article (what does this mean?). If published, this will include your full peer review and any attached files.

Reviewer #1: No

Reviewer #2: No
---

## [Editor Report · Decision Letter 1]

14 Jul 2020

Dear Dr. Kim,

We are pleased to inform you that your manuscript 'Computer-guided Binding Mode Identification and Affinity Improvement of an LRR Protein Binder without Structure Determination' has been provisionally accepted for publication in PLOS Computational Biology.

Best regards,

Marco Punta

Associate Editor

PLOS Computational Biology

Arne Elofsson

Deputy Editor

PLOS Computational Biology

---

## [Editor Report · Acceptance letter]

24 Aug 2020

PCOMPBIOL-D-20-00449R1 

Computer-guided Binding Mode Identification and Affinity Improvement of an LRR Protein Binder without Structure Determination

Dear Dr Kim,

I am pleased to inform you that your manuscript has been formally accepted for publication in PLOS Computational Biology. Your manuscript is now with our production department and you will be notified of the publication date in due course.

With kind regards,

Laura Mallard
